# Gene expression alterations in testicular biopsies from males with spermatogenesis arrest identified by transcriptome analysis

Osamah Batiha[1]*, Esra'a Al-Zoubi[1], Rowida Almomani[2], Mohammad A. Al Smadi[3], Sura Alrawabdeh[4], Omar Alshokaibi[5], Hussam Abu-Farsakh[6], Abedalrhman Alkhateeb[7], Masood Abu-Halima[8]

1 Department of Biotechnology and Genetic Engineering, Jordan University of Science and Technology, Irbid, Jordan, 2 Department of Medical Laboratory Sciences, Faculty of Applied Medical Sciences, Jordan University of Science and Technology, Irbid, Jordan, 3 Reproductive Endocrinology and IVF Unit, King Hussein Medical Centre, Amman, Jordan, 4 Consultant Pathologists, Royal Medical Services, Amman, Jordan, 5 Pathology Department, Princess Iman Research and Laboratory Sciences Center, King Hussein Medical Center (KHMC), Amman, Jordan, 6 First Medical Lab of Histopathology, Amman, Jordan, 7 Computer Science Department, Lakehead University, Thunder Bay, Canada, 8 Institute of Human Genetics, Saarland University, Homburg, Germany

☯ These authors contributed equally to this work.
* oybatiha@just.edu.jo

## Abstract

Spermatogenesis is a complex biological process encompasses several stages of cellular divisions, ultimately resulting in producing mature spermatozoa capable of fertilization. Numerous factors involved in the precise regulation of the spermatogenesis, and any disruptions or alterations in these regulatory mechanisms can lead to spermatogenesis arrest, which may result in male infertility. Among these factors, genetic influences play essential role in regulating the process. This study aimed to identify genes that are differentially expressed in relation to spermatogenesis arrest. Testicular biopsy samples were collected from 22 non-obstructive azoospermic patients diagnosed with spermatogenesis arrest (cases) and nine obstructive azoospermic patients (controls). RNA sequencing (RNA-seq) was performed on five samples from the 22 non-obstructive azoospermic patients and compared to previously published transcriptomic data from obstructive azoospermic patients, which served as the control group. Differential expression analysis of the RNA-seq data identified 1,915 differentially expressed genes, comprising 337 upregulated and 1,578 downregulated genes. Among these, several key candidate genes were identified for further analysis, including the upregulation of *FOS, FOSB, RGS1, and CXCL8*, as well as the downregulation of *TNP2, SPRR2C, LINC02314, and C16orf78*. RT-qPCR validation confirmed the RNA-seq findings for these genes in the tested samples. Subsequently, RT-qPCR was performed on the remaining 17 non-obstructive (n = 17) and obstructive azoospermic samples (n = 9) collected in this study. The results from these additional samples were consistent with the RNA-seq data, further supporting the findings. Using gene ontology

**Data availability statement:** All RNA sequence files are available from the SRA database (BioProject accession number PRJNA1233842). Samples accession number: SAMN47280701, SAMN47280702, SAMN47280703, SAMN47280704, SAMN47280705.

**Funding:** Deanship of Research at Jordan University of Science and Technology (Grant No. 2022044).

**Competing interests:** The authors have declared that no competing interests exist.

(GO) analysis and published literature, we linked these genes with spermatogenesis arrest, identifying promising targets that could serve as potential biomarkers for this condition in the future.

## Introduction

Azoospermia as defined by the World Health Organization (WHO), refers to the complete absence of sperm in the ejaculate [1]. This condition affects approximately 1% of males worldwide and 10–15% of males experiencing infertility [1]. Azoospermia is classified into two primary types: obstructive azoospermia (OA) and non-obstructive azoospermia (NOA). OA is caused by blockages in the vas deferens or seminal ducts, whereas NOA typically arises from disruptions in spermatogenesis [2,3]. Spermatogenesis is a highly synchronized, multi-phase process that leads to the formation of mature spermatozoa [4,5]. This process encompasses cell renewal, proliferation, differentiation, epigenomic remodeling, and a sequence of mitotic and meiotic divisions that produce millions of sperm in the testes [5–7]. Disruptions at any stage—whether spermatogonia, spermatocytes, or spermatids—can lead to spermatogenesis arrest (SA) and, consequently, male infertility [5,8]. Such disruptions may be influenced by genetic factors such as mutations, chromosomal abnormalities, and/or gene expression mediated by RNAs, including small RNAs and other non-coding RNAs, which are essential in the regulation of spermatogenesis and male infertility [9–14].

Recent advances in high-throughput transcriptome analysis have facilitated the discovery of numerous genes that influence semen quality and testicular histology [15]. These developments have improved our comprehension of the function of both coding and non-coding transcripts in spermatogenesis [15,16]. Research has demonstrated significant correlations between various RNAs and sperm motility, count, and, to a lesser extent, morphology [17,18]. Furthermore, transcriptomic approaches have uncovered over 60,000 transcripts within the sperm transcriptome, including approximately 11,000 that exhibit differential expression between infertile and fertile men [15,17]. Nevertheless, there exists a considerable gap in high-throughput investigations that specifically examine the coding and non-coding transcriptomes in men with azoospermia, particularly those with spermatogenic arrest at the levels of spermatogonia, spermatocytes, or spermatids [19,20].

In spite of notable advancements in elucidating the genetic and epigenetic factors that contribute to male infertility, the exact molecular mechanisms that underlie various stages of spermatogenic arrest remain inadequately understood. The majority of current diagnostic assessments depend on histological evaluations and hormonal analyses, which fail to fully represent the intricate gene expression dynamics linked to testicular dysfunction [21,22]. Consequently, thorough transcriptomic profiling—especially via RNA-sequencing (RNA-seq)—presents a promising strategy for revealing molecular signatures associated with specific arrest stages in non-obstructive azoospermia (NOA). Such revelations are essential not only for enhancing our

biological understanding of spermatogenesis but also for identifying potential diagnostic biomarkers and therapeutic targets specifically designed for NOA-related conditions.

This work utilized RNA-seq to investigate the transcriptomic profile and pinpoint differentially expressed transcripts in human testicular tissues from NOA patients at various stages of spermatogenic arrest, in comparison to control males with OA who exhibited normal testicular interstitial compartments. The findings of our results lay the groundwork for the identification of novel diagnostic biomarkers for various forms of male infertility. Additionally, our transcriptomic insights may aid in the identification of therapeutic targets for the future treatment of male infertility.

## Materials and methods

### Sample collection and processing

The study received ethical approval from the Institutional Review Board (IRB) at Jordan University of Science and Technology/King Abdullah University Hospital (Nr. 44/2022) and subsequently from the IRB at King Hussein Medical Center (Nr. 5/2022). All participants provided written informed consent prior to their inclusion in the study. Samples were collected between (22/3/2022–27/12/2023). The focus was on males diagnosed with azoospermia through semen analysis. Participants with systemic diseases, a history of chemotherapy or radiation therapy, or other infertility causes unrelated to azoospermia were excluded from the study.

Semen samples were collected on two separate occasions from a male undergoing infertility evaluation at the IVF unit of King Hussein Medical Center. Both semen analyses revealed a complete absence of spermatozoa, even after centrifugation, thereby confirming the diagnosis of azoospermia. Subjects underwent testicular examination to investigate the root cause of azoospermia and to explore potential sperm retrieval options for intracytoplasmic sperm injection (ICSI) [23]. Samples with varicocele, chronic diseases, and chromosomal abnormalities were excluded.

The testicular tissue samples were analyzed and categorized based on histopathological findings. The NOA samples were identified by SA at various stages, while OA samples displayed normal spermatogenesis, as indicated by the presence of normal sperm observed in the FNA sample during microscopic examination. A total of 31 samples were processed and included in the study, comprising 22 samples diagnosed with SA obtained through testicular sperm extraction (TESE), and 9 control samples with OA acquired via fine needle aspiration (FNA). Among the 22 SA samples, 6 were arrested at the spermatogonial stage, 3 at the spermatocyte stage, and 13 at the spermatid stage. Control samples were sourced from men with OA who demonstrated normal testicular interstitial compartments, including sperm (n = 9). The clinical characteristics of the study participants are detailed in S1 Table.

Each sample was subsequently divided into two portions: one was rapidly frozen in liquid nitrogen to maintain RNA integrity, while the other was utilized immediately for histopathological analysis in infertile cases to determine the diagnosis based on the types of cells present in each sample or cytological analysis in control samples.

Testicular tissue samples from TESE procedures that did not produce sperm underwent a histopathological examination to evaluate tissue structure and cellular characteristics. This examination employed hematoxylin and eosin (H&E) staining and adhered a methodical approach. Initially, the tissue samples were fixed in formalin, embedded in paraffin, and sectioned into thin sections. Subsequently, the slices were deparaffinized, rehydrated, and stained with hematoxylin and eosin. Following staining, the sections were dehydrated, cleared, and mounted with a coverslip for microscopic examination. Pathologists scrutinized these sections to pinpoint the specific stages of spermatogenesis and ascertain the causes of sperm absence, with particular focus on stages of spermatogenic arrest, including spermatogonia, spermatocytes, and spermatids.

RNA extraction was performed using samples preserved in liquid nitrogen (SA patients, n = 22, OA controls, n = 9). RNA isolation was conducted using Direct-zol™ RNA MiniPrep Kits (Zymo Research) following the manufacturer's protocol. In summary, approximately 25 mg of each frozen sample was weighed and homogenized with a pellet pestle in 600

µL of TRIzol™ lysis reagent and 200 µL of chloroform. After homogenization, the mixture was centrifuged to separate the phases, and the RNA-containing aqueous phase was collected. DNase I treatment was performed to prevent DNA contamination. The quality and quantity of the purified RNA samples were analysed using 1% agarose gel electrophoresis, a NanoDrop™ 2000 Spectrophotometer (Thermo Fisher Scientific), and an Agilent 2100™ Bioanalyzer (Agilent Technologies).

## RNA sequencing and analysis

Five patient samples from individuals with SA -one at the spermatogonial stage and four at the spermatid stage- were sent to Macrogen Inc. (Seoul, Republic of Korea) for RNA-seq using the Illumina NovaSeq6000 System paired-end sequencing technology. A quality control test was performed, and only samples with an RNA Integrity Number greater than 4.2 and a concentration exceeding 100 ng/µL were used for sequencing.

The control samples obtained from the IVF clinic for RNA-seq were omitted due to inadequate RNA concentrations, a limitation attributed to the use of FNA for sample collection. Consequently, five published datasets from males with OA who had undergone vasectomy reversal (as detailed in S2 Table) were incorporated for control comparisons, ensuring they were analysed using the same technology employed in our study [24]. Comprehensive details for these control samples can be found in S2 Table.

The RNA sequences generated in this project have been submitted to the SRA database [https://www.ncbi.nlm.nih.gov/sra/?term=PRJNA1233842] (BioProject accession **PRJNA1233842**). The accession numbers of RNA-seq data of all used samples are represented in S3 Table.

The RNA-seq data were evaluated by comparing the SA patients (n = 5) with the previously published OA controls (n = 5), with an emphasis on identifying differentially expressed genes (DEGs). The raw data from the sequencing processes, which include transcriptome sequencing data, underwent several processing steps. These analyses were performed using the European Galaxy server (https://usegalaxy.eu/).

The raw data of SA samples were acquired from the Macrogen Inc. sequencing facility, and the OA raw data were downloaded from the Sequence Read Archive (SRA) database. These data were converted to FASTQ format using the "Convert SRA to FASTQ" tool. Initial quality assessment was performed using FastQC (v0.11.9) and MultiQC (v1.10.1). On average, OA control samples had 32,852,033, 36,150,024, 43,314,658, 42,104,473, and 43,784,740 clean reads, while SA patient samples had 24,430,153, 21,001,865, 23,644,940, 19,973,446, and 20,631,411 clean reads, respectively. Adapter sequences were removed, low-quality bases were trimmed, and short reads were filtered using the Trimmomatic tool (v0.39). The quality of the trimmed data was reassessed with FastQC and MultiQC to ensure data integrity. The trimmed reads were then aligned to the reference genome (GRCh38.p14) using the RNA STAR aligner (v2.7.9a). The percentage of successfully aligned sequences exceeded 80% for all samples, as indicated by the MultiQC report derived from the RNA STAR log files. Read counts per annotated gene were quantified using the FeatureCounts tool (v2.0.1). After generating the matrix containing read counts for each sample, including SA patients and OA controls, differential expression analysis was conducted using DESeq2 (v1.34.0). This tool normalized the read counts and identified DEGs. The DEGs underwent filtration through two steps, beginning with the exclusion of genes with p-value <0.05, followed by the removal of genes with log2FC between (−2 and +2). Samples were then sorted in descending order based on log2FC, and genes with the highest log2FC and the lowest log2FC with a significant p-value were selected for validation using RT-qPCR.

Subsequently, R software (v4.2.1, R Core Team) was used for further analysis and visualization of the results, including heatmaps, volcano plots, Principal Component Analysis (PCA), Spearman's rank correlation coefficients, and other visualizations. Differentially expressed transcripts were selected based on two criteria: a Log2 fold change (Log2FC) greater than 2 (indicating upregulation) or less than −2 (indicating downregulation), and an adjusted p-value using the Benjamini and Hochberg method [25].

Gene Ontology (GO), encompassing Cellular Components (CC), Molecular Function (MF), and Biological Processes (BP), along with Kyoto Encyclopedia of Genes and Genomes (KEGG) pathway analyses, was utilized to identify enriched

biological pathways and gene ontologies. This approach emphasizes the functional roles and pathways related to differentially expressed genes. The biological pathways and processes were illustrated using the enrichplot package from the clusterProfiler tool [26] in R, which facilitated the interpretation of enrichment results. An adjusted p-value threshold of < 0.05 was applied for the analysis.

### Reverse transcription quantitative polymerase chain reaction (RT-qPCR)

To validate the RNA-seq results, RT-qPCR was performed. The relative quantification method was employed to assess changes in gene expression levels. After conducting RNA-seq analysis, we selected the top four up-regulated and top four down-regulated genes for validation through quantitative PCR (qPCR) on SA patient (n = 22) and OA control (n = 9) samples. Reverse transcription polymerase chain reaction (RT-PCR) was first performed using the Mir-X™ miRNA First-Strand Synthesis and TB Green™ qRT-PCR Kit (Takara Bio USA). The reaction mixture consisted of 5 µL of mRQ buffer (2X), 1.25 µL of mRQ enzyme, 0.25 µg of RNA sample, and nuclease-free water, for a total volume of 10 µL. The reaction was incubated at 37°C for 1 hour, followed by enzyme inactivation at 85°C for 5 minutes according to the manufacturer's protocol. To prepare the cDNA sample, 90 µL of nuclease-free water was added, bringing the total volume to 100 µL.

Subsequently, qPCR was conducted using the prepared cDNA samples and the TB Green™ Premix Ex Taq™ II (Tli Ranesh Plus) Kit (Takara Bio USA). The PCR reaction was prepared by adding 5 µL of 2X master mix, 1 µL of each primer, 1 µL of cDNA, and 2 µL of nuclease-free water to achieve a final reaction volume of 10 µL. The thermal cycling conditions were as follows: an initial denaturation at 95°C for 15 minutes, followed by 40 cycles of three steps—denaturation at 95°C for 15 seconds, annealing at 55°C for 30 seconds for *TNP2, LINC02314, SPRR2C, C16orf78,* and *CXCL8* genes, and 60°C for 30 seconds for *FOS, FOSB,* and *RGS1* genes and an extension at 70°C for 30 seconds. The run concluded with a melting curve analysis involving a gradual temperature increase with the following steps: 95°C for 1 minute, 60°C for 15 seconds, and 72°C for 1 minute, with continuous monitoring of fluorescence. Each sample was run in duplicate, with a no-template negative control included in each run. *GAPDH* was used as an internal control for normalizing the Ct values, and relative gene expression levels were calculated using the ΔΔCt method.

The PCR products were resolved on a 2% agarose gel in 1X TBE buffer (89 mM Tris, 89 mM boric acid, 2 mM EDTA), stained with ethidium bromide, and electrophoresed at 120 V for 30 minutes. Bands were visualized using a UVP GelDoc-It 2 Imaging System (Fisher Scientific) with a 50 bp DNA ladder as reference. Primer sequences used for qPCR are listed in S4 Table.

Finally, the ΔCt value, calculated as the difference between the Ct value of the gene of interest and the Ct value of the endogenous reference gene, was determined using the formula: ΔCt = Ct (gene of interest) − Ct (endogenous reference gene). *GAPDH* was selected as the endogenous reference gene for normalization. RT-qPCR was performed in duplicates for each sample, and statistical significance was assessed using an unpaired two-tailed t-test, with a p-value of < 0.05 was considered significant.

## Results

### Clinical and histological evaluation of the patient cohort

The clinical features of the SA patients (n = 22) and the OA controls (n = 9) are detailed in S1 Table. SA patients at various stages of spermatogenic arrest exhibited significantly altered levels of testosterone and prolactin compared to control men (p < 0.05). However, other parameters, such as age, luteinizing hormone, and follicle-stimulating hormone, did not show significant differences.

Testicular samples were histologically analyzed, as illustrated in Fig 1. The study included testicular tissues arrested at the spermatogonial stage (n = 6), the spermatocyte stage (n = 3), and at the spermatid stage (n = 13). Additionally, control samples from men with obstructive azoospermia (OA) exhibiting normal testicular interstitial compartments (n = 9) were included.

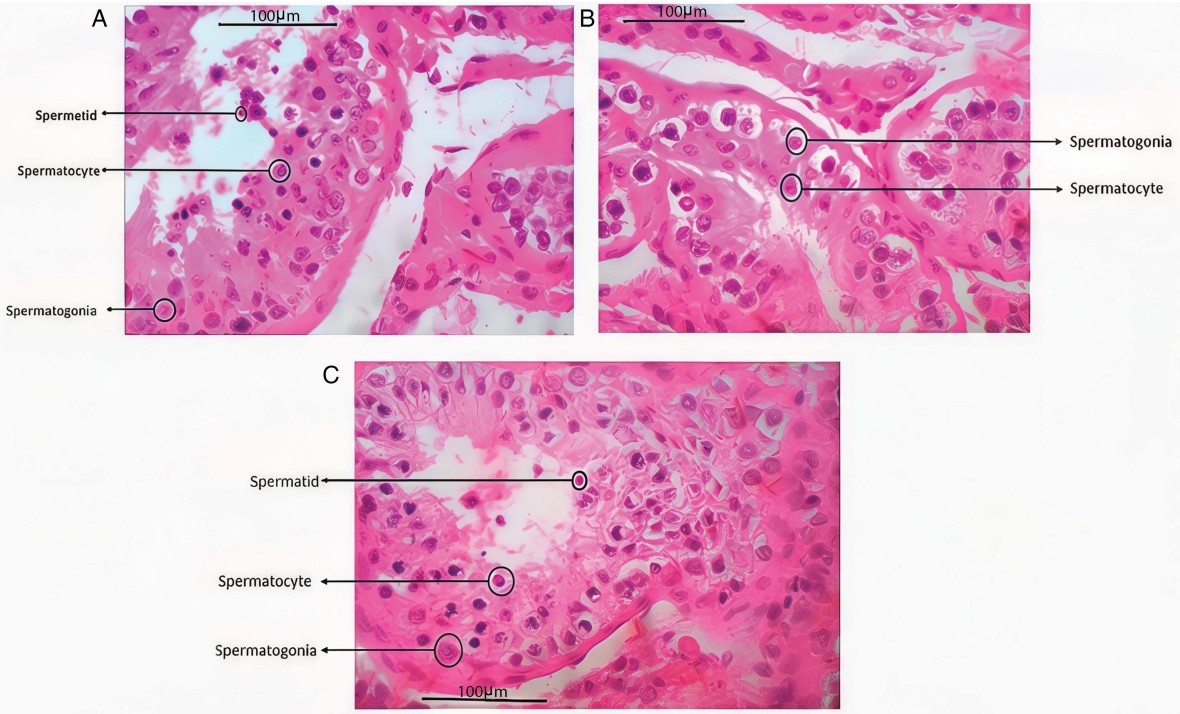

**Fig 1. Histopathological analysis of three distinct testicular samples.** Panels A and C depict samples arrested at the spermatid stage, while Panel B represents a sample arrested at the spermatocyte stage.

### RNA-Seq data analysis and validation

RNA-seq was performed on a smaller subset of samples (n = 5 SA, n = 5 OA) for high-throughput analysis. Differential expression analysis was conducted to identify transcripts exhibiting varying expression levels in testicular tissue samples from patients at different stages of spermatogenic arrest (SA, n = 5) in comparison to controls with (OA, n = 5). After estimating variance-mean dependence, conducting differential expression testing, and normalization using DESeq2, expression levels for each group (SA and OA) were established. By adjusting for multiple comparisons using the Benjamini-Hochberg method and focusing on transcripts with a fold change of ≥ 2 in either direction, 1,915 transcripts with significant expression differences in the SA group relative to the OA controls were identified (adjusted p < 0.05, fold change ≥ 2). Among these, 1,578 transcripts were significantly down-regulated (adjusted p-value <0.05, fold change ≤ −2), while 337 were significantly up-regulated (adjusted p < 0.05, fold change ≥ 2). Of the identified differentially expressed transcripts, 626 were protein-coding genes, with 409 down-regulated and 217 up-regulated (adjusted p < 0.05, fold change ≥ 2). The results were validated using the top four down-regulated genes (*TNP2, SPRR2C, LINC02314, and C16orf78*) and the top four up-regulated genes *(FOS, FOSB, RGS1, and CXCL8)* that were revealed through this analysis. A volcano plot (Fig 2A) visualizes the differential expression of protein-coding genes, with log2 fold-change plotted against the − log10 adjusted p-value, comparing SA patients to OA controls.

To assess the extent of correlation among individual samples within each group, i.e., SA patients and OA controls, pairwise sample correlations were computed using Spearman's correlation coefficient based on gene expression levels present in the considered sample group. The correlation between the samples was higher within each group compared to between groups (Fig 2B). The dendrograms on the top and left sides of the heatmap represent hierarchical clustering of samples and features, respectively, highlighting distinct gene expression patterns between the SA patients and OA

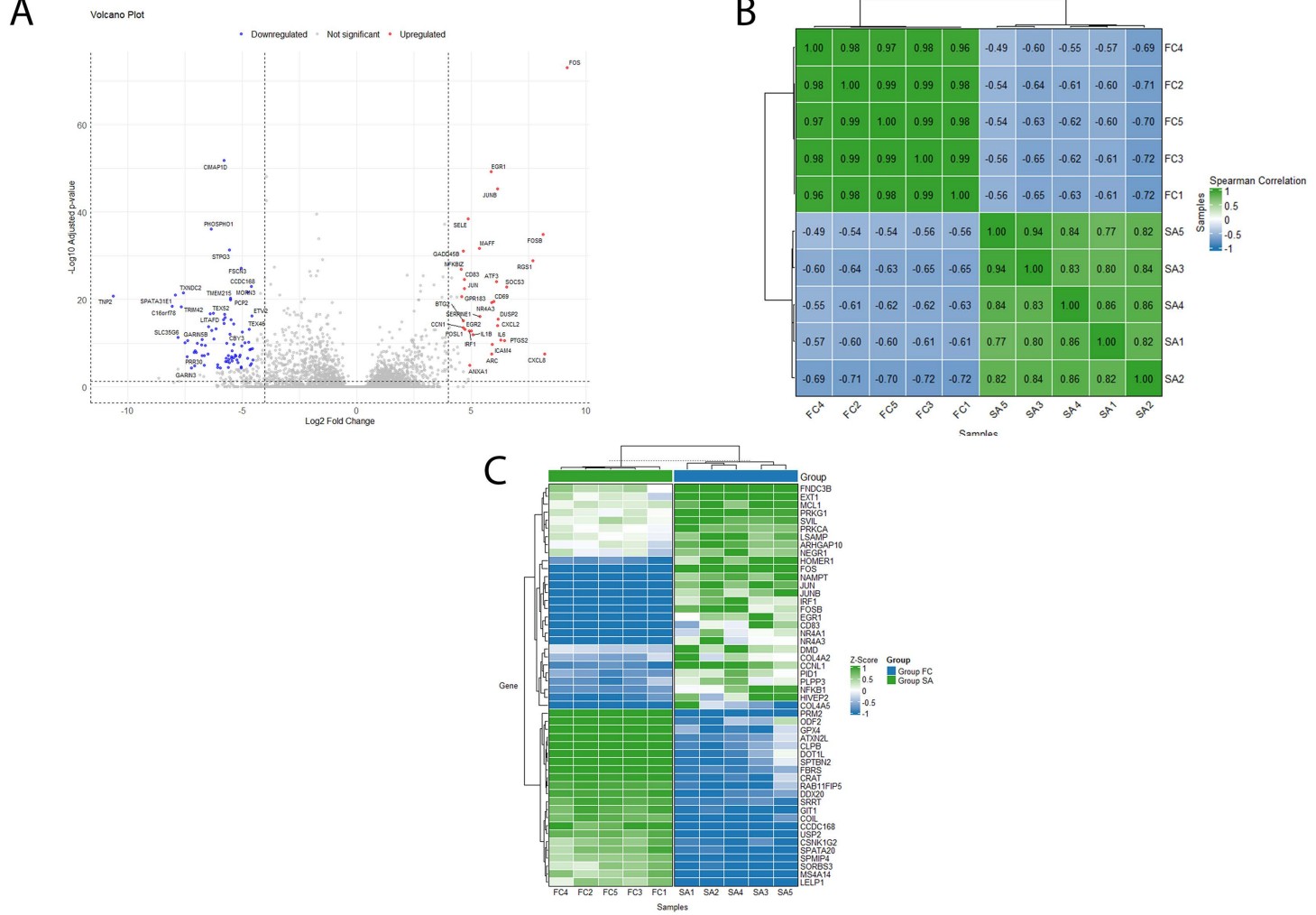

**Fig 2. Differential Gene Expression Profiles in spermatogenesis arrest (SA) and obstructive azoospermic (OA) Cohorts.** (A) Volcano plot illustrating the differential gene expression between spermatogenesis arrest (SA) patients and obstructive azoospermia (OA) controls. Genes that are up-regulated in SA patients are shown on the right, while down-regulated genes are depicted on the left. The plot is based on log2 fold change (log2FC) and adjusted p-values. (B) Heatmap representing the pairwise Spearman's correlation coefficients for gene expression profiles of samples from SA patients and OA controls. (C) Heatmap showing the variation in gene expression levels between SA patients and OA controls. The colors indicate Z-scores, which represent deviations from the mean gene expression across all samples.

controls. The correlation analysis indicated that the samples within each category exhibited higher correlations with one another compared to samples from different categories.

To investigate the correlation between SA patients and OA controls in terms of expression levels, hierarchical clustering was performed using the protein-coding genes with the highest expression variance among the identified genes. The resulting heatmap (Fig 2C) reveals two distinct, non-overlapping clusters: one comprising only SA patients and the other only OA controls. This heatmap illustrates that certain differentially expressed protein-coding genes were uniquely expressed in the SA group or at low levels in the OA group, and vice versa (Fig 2C).

To validate our RNA-Seq findings, we performed RT-qPCR analyses on eight selected genes ((*TNP2, SPRR2C, LINC02314, C16orf78*), (*FOS, FOSB, RGS1, and CXCL8*)). The initial validation utilized the same SA samples from the

RNA-Seq study (n = 5). Subsequently, the cohort was expanded to include 28 samples: SA (n = 17) and OA samples (n = 9). Genes were chosen based on a fold change ≥ 2, adjusted p < 0.05) and biological relevance to spermatogenesis (Fig 3A).

The PCA analysis of the dataset indicating a clear distinction between SA patients and OA controls, based on the first two principal components (PC1 at 87.3% and PC2 at 8.3%) accounting for the variance (Fig 3A). The RT-qPCR results supported the RNA-seq data, confirming the direction of expression changes for all eight genes, with minor differences in fold change magnitude. In SA patients, *FOS, FOSB, RGS1,* and *CXCL8* were significantly upregulated, showing an increase of 9.67-fold, 5.04-fold, 4.59-fold, and 4.08-fold, respectively, compared to OA controls (Fig 3A). Conversely,

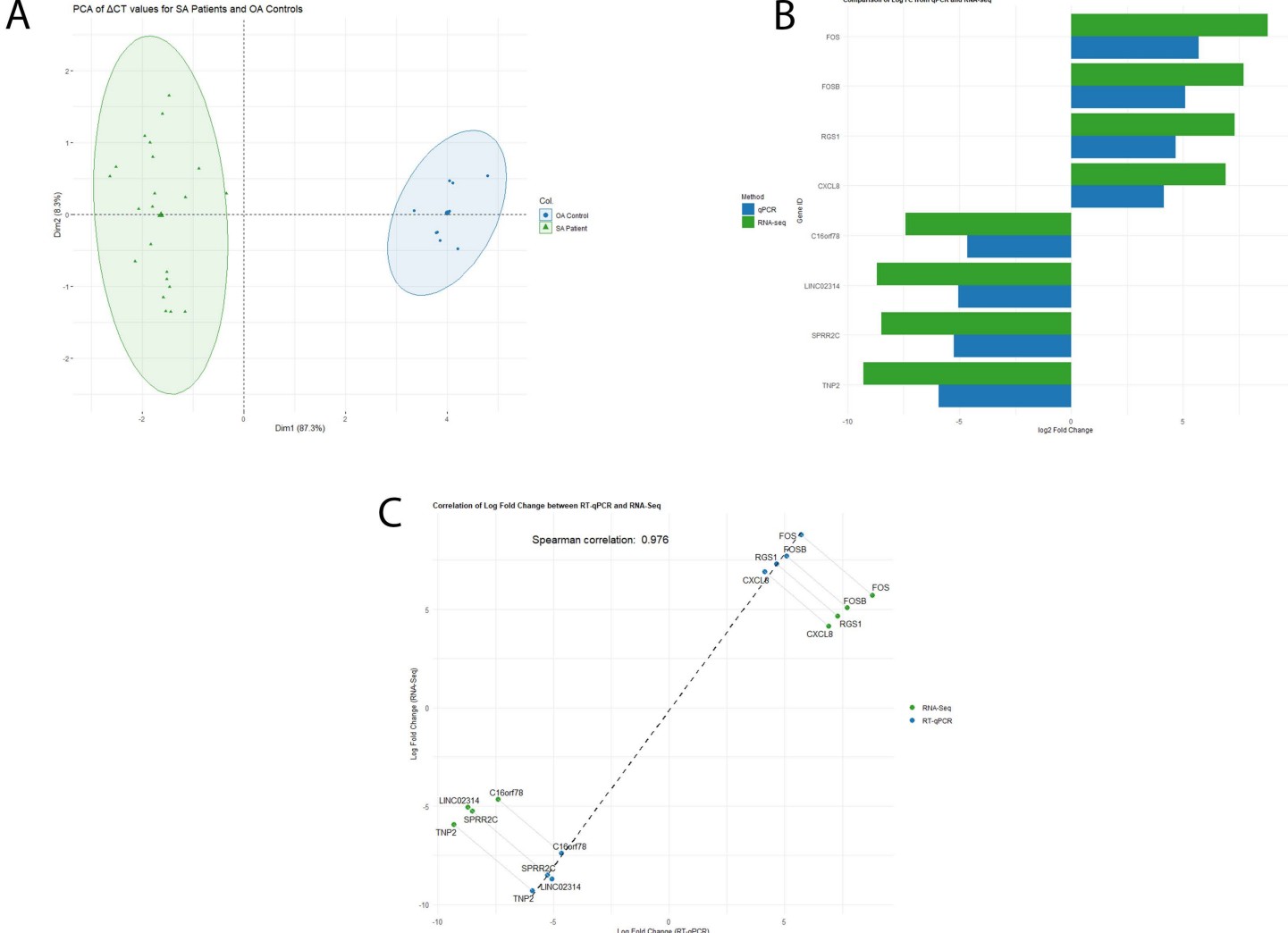

**Fig 3. Transcriptomic Profiling and qPCR Validation of Differentially Expressed Genes in spermatogenesis arrest (SA) and obstructive azoospermic (OA) Groups.** (A) Principal Component Analysis (PCA) of gene expression profiles comparing SA patients and OA controls. The separation of samples into two distinct groups highlights the differential gene expression between the cohorts. (B) Differential expression analysis of the top four up-regulated and down-regulated genes validated using qPCR. Data are presented as log2FC to demonstrate the extent of expression differences. (C) A comparative graph shows the expression levels of the top four up-regulated and down-regulated genes across RNA-seq and qPCR methods, highlighting the consistency and discrepancies between the two approaches.

*TNP2, SPRR2C, LINC02314*, and *C16orf78* were markedly downregulated, with decreases of −5.9-fold, −5.06-fold, −5.26-fold, and −4.67-fold respectively (Fig 3B). A Spearman's rank correlation coefficient of 0.976 between RT-qPCR and RNA-seq data underscores the reliability and accuracy of our findings (Fig 3C). This high degree of concordance highlights the significant differential expression of these genes in SA patients relative to OA controls.

## Functional and pathway enrichment analysis

The Gene Ontology (GO) enrichment analysis of differentially expressed genes from the comparison between the SA samples and the five OA samples revealed a strong association with the Biological Process (BP) category, which had the highest number of significant subterms (254). In contrast, the Molecular Function (MF) analysis identified 62 significant subterms, and the Cellular Components (CC) analysis found 64 significant subterms. The most significant subterms related to spermatogenesis from each main category are shown in Fig 4A. Additionally, KEGG analysis was performed to pinpoint the most significantly impacted biological pathways among the DEGs from the comparison of the 5 FC and 5 SA samples. Pathways were considered significant with a p-value threshold of <0.05. The results, displayed in Fig 4B, highlight 10 significant pathways ranked by the number of DEGs involved.

## Discussion

Spermatogenesis arrest is a multifaceted condition that can manifest at various stages of sperm development, influenced by both genetic and environmental factors [8]. The precise regulation of gene expression in germ cells is essential for normal spermatogenesis and subsequent male fertility.

In this study, we utilized high-throughput RNA sequencing and bioinformatics analyses to explore the transcriptome landscape of SA in infertile males with NOA. Our analysis revealed 1,915 DEGs, including 337 that were upregulated and 1,578 that were downregulated. GO enrichment analysis indicated that these DEGs are closely associated with spermatogenesis and various reproductive functions. Among the upregulated genes, *FOS, FOSB, RGS1*, and *CXCL8* were particularly notable, whereas *TNP2, SPRR2C, LINC02314*, and *C16orf78* were significantly downregulated. Validation using RT-qPCR confirmed their differential expression, underscoring their potential roles in cellular stability, stress response, and structural regulation.

Pathway analysis further suggested that these genes are implicated in critical biological processes, including the cell cycle, Mitogen-Activated Protein Kinase (MAPK) signaling, and p53 signaling pathways, all of which have substantial implications for male infertility. Notably, the chromatin condensation process during spermatogenesis is a two-step event involving the temporary replacement of testis-specific histones by transitional proteins (*TNP1* and *TNP2*), followed by protamines (*PRM1* and *PRM2*) [27,28]. These proteins are essential for maintaining the integrity and stability of RNA and DNA.

During prophase I of meiosis, DNA double-strand breaks take place as part of the homologous recombination process. *TNP1* and *TNP2* play a crucial role in repairing these breaks, which is essential for preserving genetic stability during spermatogenesis [29,30]. These discoveries provide new perspectives on the molecular mechanisms that contribute to male infertility and highlight the importance of these genes in reproductive health.

Transition protein-2, a nuclear transition protein involved in critical processes such as single fertilization, spermatogenesis, and sperm binding to the zona pellucida. Its pivotal role in DNA condensation is essential for the proper formation and maturation of sperm [30–34]. Dysregulation of the *TNP2* gene can lead to abnormal DNA packaging and chromatin structure [30,35], resulting in impaired sperm morphology and function, which may obstruct sperm binding to the zona pellucida and the successful completion of fertilization. Furthermore, *TNP2* is linked to zinc ion binding, a process crucial for enhancing sperm fertilization capability by improving motility and capacitation [36]. Zinc is vital for stabilizing the sperm mitochondrial sheath and chromatin [37–40], and its deficiency has been correlated with reduced male fertility. *TNP2* interacts with zinc ions through arginine- and cystine-rich protamines, which assist in stabilizing protein structure

**A**

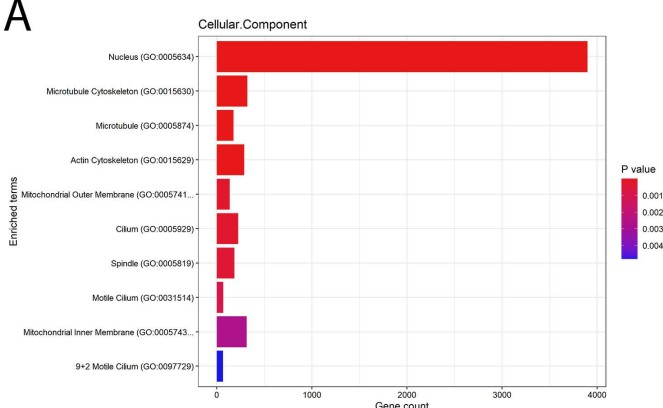

**B**

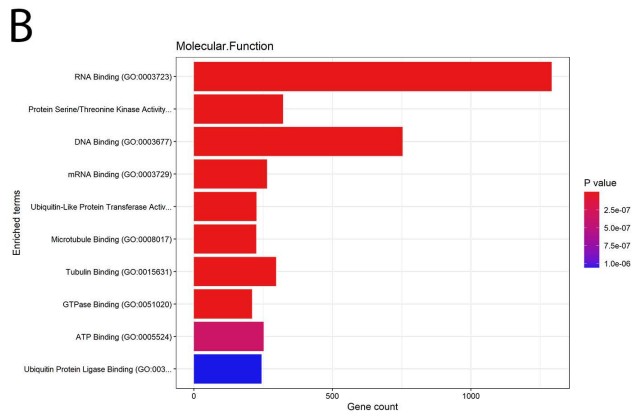

**C**

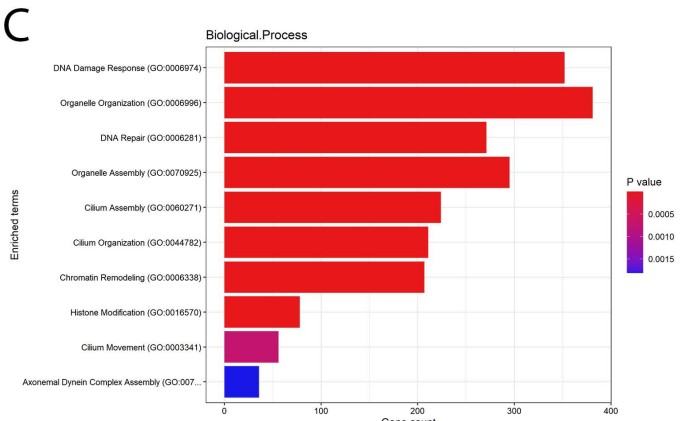

**D**

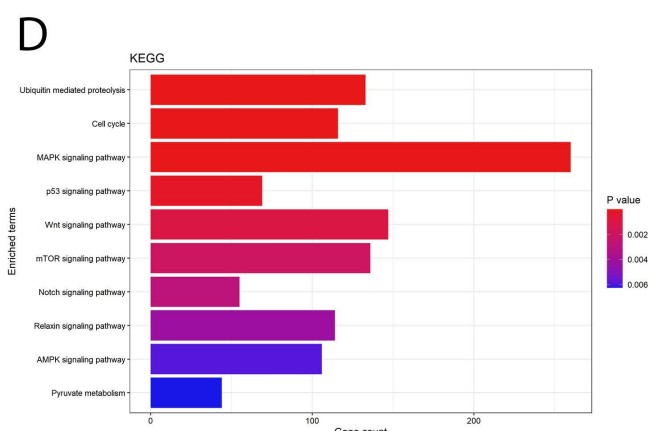

**Fig 4. Gene Ontology (GO) and Kyoto Encyclopedia of Genes and Genomes (KEGG) pathway enrichment analyses were performed on the significantly differentially expressed genes.** Results are shown as follows: (A) Cellular Components (CC), (B) Molecular Function (MF), (C) Biological Processes (BP), and (D) KEGG pathways. The gene ratio on the x-axis represents the proportion of input genes associated with each term, relative to the total annotated genes in that category. The size of each dot corresponds to the number of genes associated with the term, while the color gradient in the bar chart represents each pathway's p-value, with red indicating higher significance (lower p-values) and blue indicating lower significance (higher p-values within the range). The X-axis shows the number of genes associated with each pathway's enrichment.

and chromatin remodeling during spermatogenesis [36]. However, elevated zinc levels can induce mitochondrial oxidative stress due to its pro-oxidant properties [41]. Genetic variations in the *TNP2* gene are associated with defects in sperm quality and quantity, potentially contributing to SA due to its critical role in chromatin condensation [31,32,42,43].

In this work, we observed a notable decrease in *TNP2* expression within the SA group, aligning with earlier studies that indicate reduced *TNP2* levels in individuals diagnosed with NOA, including SA and Sertoli cell-only syndrome (SCOS) [32,33,43–45]. Notably, while diminished *TNP2* expression is associated with impaired sperm quality, elevated *TNP2* expression has been linked to abnormal sperm morphology and teratozoospermia [31]. These findings underscore the importance of maintaining *TNP2* expression for optimal spermatogenesis and favorable fertility outcomes.

The progression of germ cells from the spermatogonial stage to the production of mature sperms is influenced by several pathways, including the MAPK pathway [46]. This pathway is essential in the regulation of spermatogenesis, as

it facilitates the proliferation of Sertoli cells and spermatogonial germ cells, in addition to promoting germ cell growth and apoptosis.

In this study, we have identified *FOS*, *FOSB*, and *CXCL8* as significantly upregulated genes, all recognized as downstream targets of MAPKs and potentially implicated in SA. *FOS* and *FOSB*, which are part of the activator protein-1 (AP-1) transcription factor family, play a role in regulating cell proliferation, differentiation, and survival via the MAPK pathway [47,48].

Interestingly, our study observed elevated expression of *FOS* and *FOSB* in the SA group, while a previous study reported a decrease in expression of these genes among infertile men [47]. This discrepancy may arise from differences in study design and sample characteristics. Our study included cases arrested at various spermatogenic stages, from spermatogonia to spermatids, whereas the previous study mainly focused on maturation arrest and SCOS cases [47]. Additionally, the previous study utilized immunohistochemistry to measure *c*-FOS protein expression without distinguishing between its phosphorylated and non-phosphorylated forms. Their findings showed predominantly negative expression in germ and somatic cells, with positive expression limited to type B spermatids [47]. In contrast, our study employed gene expression analysis, demonstrating elevated *c*-FOS expression in NOA cases compared to OA. These differences likely reflect variations in sample composition and methodology. The previous study included samples lacking germ cells, such as those from SCOS, and those with spermatid stage arrest. Whereas our study focused exclusively on SA cases, covering a spectrum from the spermatogonial to the spermatid stage. This distinction in sample types likely played a significant role in the observed variation in gene expression patterns.

The C-X-C motif chemokine ligand 8 (*CXCL8*) gene encodes the interleukin-8 (IL-8), a chemokine protein, that is vital for the activation of the MAPK pathway. Chemokines, along with cytokines, are essential for facilitating intracellular communication, particularly in mediating inflammatory responses [49]. IL-8is secreted by various immune cells, including mononuclear cells, T lymphocytes, and macrophages, as part of the body's defense mechanism against infection. Its function involves binding to G-protein-coupled receptors, leading to the release of beta and gamma G-protein subunits that activate downstream pathways such as MAPK. Additionally, *CXCL8* contributes to microtubule cytoskeleton reorganization, thereby enhancing the migration of the immune cells [50]. Our findings revealed that *CXCL8* was significantly overexpressed in the SA group compared to the FC group, consistent with previous studies linking elevated *CXCL8* expression with heightened production of reactive oxygen species (ROS). Excessive ROS levels induce DNA fragmentation in sperm, which hinders sperm functionality and contributes to male infertility [51]. These findings suggest that while *CXCL8* plays a protective role in immune activation and signaling pathways, its overexpression may negatively impact fertility through ROS-induced DNA damage.

In similar manner, the regulator of G protein signaling 1 (*RGS1*) gene exhibited elevated expression levels in the SA group. RGS1 belongs to the regulator of G-protein family and is crucial in modulating G-protein-coupled receptor (GPCR) signaling pathways [52], which are vital for cellular responses to extracellular stimuli such as hormones and chemokines [53]. The RGS1 protein encoded is localized on the cytosolic side of the plasma membrane and contains a specialized RGS domain. Although no direct evidence links *RGS1* to SA, its potential role in spermatogenesis has been proposed. Spermatogenesis involves a complex interaction of signaling pathways, including GPCR pathways, that are essential for the proper functioning of germ cells, Sertoli cells, and Leydig cells [54]. *RGS1* facilitates the deactivation of G-proteins, a critical process for regulating GPCR-mediated actions [52]. A defect in *RGS1* could disrupt the GPCR pathway, consequently affecting processes governed by this pathway, including spermatogenesis. The *RGS1* gene's involvement in GPCR signaling highlights its importance in regulating pathways that may be critical for spermatogenesis and male fertility.

The *LINC02314* gene, classified as a long non-coding RNA (lncRNA), has been observed to be downregulated in patients with SA [55]. This gene is recognized for its role in regulating both transcriptional and post-transcriptional gene expression, as well as its involvement in RNA and DNA binding. Despite direct evidence linking *LINC02314* to SA is

limited, it is high expression in testicular tissues implies that it may influence key regulatory mechanisms, such as chromatin remodeling, transcriptional regulation, and post-transcriptional processing [55]. These processes are vital for proper chromatin condensation and spermatogenesis in germ cells.

A small proline-rich protein-2C (*SPRR2C*), a pseudogene product linked to the development of the cornified epithelial cell envelope (CE), has been detected in testicular tissue, indicating a potential role in testicular epithelial cells [56,57]. The CE serves as a protective barrier [56,57], while epithelial cells in the testis are essential for sperm maturation, storage, and preventing autoimmune responses [58]. Specifically, Sertoli cells, specialized epithelial cells located within the seminiferous tubules, provide structural and nutritional support to germ cells and regulate hormonal activity during spermatogenesis [58,59]. The presence of SPRR2C in testicular tissues indicates its possible involvement in maintaining the epithelial environment that is essential for spermatogenesis and the overall health of sperm.

The *C16orf78* gene, found to be downregulated in SA patients, is currently unannotated and has one known transcript. Its precise function remains unclear, however, transcriptome analyses indicate that its expression is predominantly restricted to testicular tissue among various human organs [60]. This tissue- specific expression suggests a potential involvement in spermatogenesis or other testicular functions, necessitating additional research to clarify its exact role.

This study has several limitations. One important point is the small sample size used in the RNA-seq analysis (n = 5 per group), which inherently reduces statistical power and increases the risk of false positives or negatives. This limitation arose from several unavoidable constraints, including the unavailability of informed consent from all biopsy patients, histopathological exclusions, and the need to preserve some biopsy material for IVF use.

Moreover, some samples failed RNA or cDNA quality control thresholds. To overcome these difficulties, we used RNA-seq as a discovery tool and complemented it with RT-qPCR validation in an expanded, independent cohort (n = 22 SA, n = 9 OA), which significantly strengthened the reliability of our findings. Another key limitation is the inconsistency in the sources of control samples: RNA-seq controls were obtained from a public dataset due to technical failure in local control RNA preparations, while RT-qPCR controls were locally sourced. This raises the possibility of ethnic or technical variability. We mitigated this by carefully selecting a public dataset with matched clinical features and sequencing protocols, and emphasized validation using local samples. Additionally, the RT-qPCR validation cohort was not stratified by arrest stage due to limited sample sizes, which may obscure stage-specific expression patterns. Nonetheless, we performed post hoc subgroup analyses of the validated genes across different SA stages (data are not shown), which revealed no statistically significant differences. The number of validated genes (eight in total) was also constrained by cDNA yield, especially in low-quality clinical samples, though we applied strict statistical ($p < 0.05$ and $\log_2FC > \pm 2$) and biological relevance criteria to prioritize gene selection. Another limitation is the absence of Y chromosome microdeletions screening prior to sample inclusion, which may confound transcriptomic results. Lastly, although efforts were made to harmonize sequencing conditions between in-house and public datasets, the risk of batch effects cannot be entirely ruled out.

Despite these challenges, the study successfully identified biologically relevant DEGs and dysregulated pathways (e.g., *MAPK, p53, GPCR*), using an integrative approach combining RNA-seq, RT-qPCR, and functional enrichment analyses. Notably, genes such as *TNP2, CXCL8, FOS, FOSB,* and *RGS1* emerged as potential diagnostic or therapeutic targets. These findings lay important groundwork for future research into the molecular mechanisms of spermatogenic arrest and the development of clinically useful biomarkers or treatment strategies for non-obstructive azoospermia.

## Conclusion

In conclusion, our study emphasizes the potential involvement of certain genes in SA, either through direct effects on germ cells or by influencing the microenvironment, such as the function of Sertoli cells. RNA-seq analysis has revealed numerous genes potentially associated to this condition, including several that may serve as stage-specific markers of arrest. From this extensive dataset, the association of eight genes with SA was validated through qRT-PCR. Nevertheless,

additional functional studies are essential to ascertain the precise roles of these genes and to deepen our understanding of their genetic contributions in the spermatogenesis process.

## Supporting information

**S1 Table. Clinical Characteristics of Included Subjects.** Control samples with OA, n = 9, Spermatogenic arrest (SA), n = 22, Mean values were reported with their corresponding standard deviation of the mean (SD), Statistical significance was determined using a non-parametric Mann–Whitney U test, with a significance threshold set at P < 0.05.
(DOCX)

**S2 Table. Details of Control Samples from Published Datasets of Obstructive Azoospermic Males Following Vasectomy Reversal.** S2 Table provides comprehensive details regarding the control samples used in the comparison with our SA samples, including chromosomal analysis results, and diagnosis.
(DOCX)

**S3 Table. Accession number of RNA-seq data.** S3 Table represents the BioProject accession number corresponding to our samples, together with the accession numbers of each sample.
(DOCX)

**S4 Table. qPCR Primer Sequences and Annealing Temperatures.** S4 Table shows the qPCR primers used to assess the expression levels of each gene. These primers were designed in the exon-exon junctions to inhibit binding to DNA sequences.
(DOCX)

## Acknowledgments

We gratefully acknowledge the assistance of the Histopathology department team at Prince Iman Center for research and laboratory and the IVF center at King Hussein Hospital for their help in sample collection and diagnosis. We would like to thank Dr. Khaldon Bodoor for his valuable contribution in editing the manuscript and for his insightful suggestions that helped to improve the quality of the work.

## Author contributions

**Conceptualization:** Osamah Batiha, Rowida Almomani.

**Data curation:** Esra'a Al-Zoubi, Sura Alrawabdeh, Omar Alshokaibi.

**Formal analysis:** Abedalrhman Alkhateeb, Masood Abu-Halima.

**Funding acquisition:** Osamah Batiha.

**Investigation:** Esra'a Al-Zoubi, Masood Abu-Halima.

**Methodology:** Osamah Batiha, Esra'a Al-Zoubi.

**Project administration:** Osamah Batiha, Esra'a Al-Zoubi, Rowida Almomani.

**Resources:** Osamah Batiha, Mohammad A. Al Smadi, Sura Alrawabdeh, Omar Alshokaibi.

**Software:** Abedalrhman Alkhateeb.

**Supervision:** Osamah Batiha.

**Validation:** Esra'a Al-Zoubi.

**Visualization:** Esra'a Al-Zoubi, Hussam Abu-Farsakh.

**Writing – original draft:** Osamah Batiha, Esra'a Al-Zoubi.

**Writing – review & editing:** Osamah Batiha, Rowida Almomani, Masood Abu-Halima.

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
