## [Decision Letter · Decision Letter 0]

3 Jul 2025

Dear Dr. Batiha,

The comments made by reviewer No. 1 regarding methodological issues that should be discussed and addressed in a convincing manner are particularly concerning.

We look forward to receiving your revised manuscript.

Kind regards,

Joël R Drevet, Ph.D.

Academic Editor

PLOS ONE

Journal Requirements:

“Deanship of Research at Jordan University of Science and Technology (Grant No. 2022044).”

Additional Editor Comments:

Dear Authors,

Two reviewers with extensive expertise in the field felt that your report could be of interest to PLoS ONE readers, provided that it is thoroughly revised in accordance with the detailed comments in the reviews below. Please respond specifically to each of the reviewers' comments and do not limit yourself to superficial changes, as only one revision will be granted.

Reviewers' comments:

Reviewer's Responses to Questions

**Comments to the Author**

1. Is the manuscript technically sound, and do the data support the conclusions?

Reviewer #1: Partly

Reviewer #2: Yes

2. Has the statistical analysis been performed appropriately and rigorously?

Reviewer #1: Yes

Reviewer #2: Yes

3. Have the authors made all data underlying the findings in their manuscript fully available?

Reviewer #1: Yes

Reviewer #2: Yes

4. Is the manuscript presented in an intelligible fashion and written in standard English?

Reviewer #1: Yes

Reviewer #2: Yes

Reviewer #1: Summary of the Study

The authors conducted a comparative transcriptomic study to identify differentially expressed genes in non-obstructive azoospermic men with spermatogenesis arrest, compared to obstructive azoospermic (OA) men with normal spermatogenesis. They performed RNA sequencing on testicular tissue biopsies (n=5 SA, n=5 OA) and validated selected DEGs via RT-qPCR in an expanded sample cohort (n=22 SA, n=9 OA). They found 1,915 DEGs (337 upregulated, 1,578 downregulated), identifying eight key genes (FOS, TNP2, CXCL8) potentially involved in SA. Functional enrichment (GO and KEGG) analyses highlighted pathways such as MAPK signaling and p53 signaling as critical in spermatogenesis regulation.

Merits

• Novelty and Relevance: The study targets a significant clinical problem—male infertility due to spermatogenic arrest. By focusing on NOA with testicular histological staging, the authors address an underexplored but critical area.

• Methodological Rigor:

1. RNA-seq quality control, alignment (STAR), and statistical analysis (DESeq2, Benjamini-Hochberg) were correctly implemented.

2. External public datasets were used as OA controls due to RNA quality issues, which is a practical solution given sample constraints.

3. RT-qPCR validation across a broader sample base strengthens the findings.

• Comprehensive Bioinformatics Analysis: Functional annotations (GO/KEGG), PCA, volcano plots, and correlation matrices provide strong visual and statistical support for the findings.

• Data Accessibility: Raw sequencing data has been deposited in the SRA (BioProject PRJNA1233842), fulfilling PLOS ONE’s open data policies.

3. Limitations and Areas for Improvement

A. Major Concerns

1. Small Sample Size for RNA-Seq (n=5 SA): This is a serious limitation for robust differential expression analysis. Statistical power is restricted, potentially inflating false positives/negatives.

2. Control Sample Source Inconsistency: The OA samples used for RNA-seq were from a public dataset, while qPCR controls were locally collected. This raises concerns of ethnic variability, and technical heterogeneity despite efforts to align sequencing protocols. If authors do not have to do with this, then they should consider it as major limitation of their study.

3. Gene Selection Justification: The basis for selecting the eight genes (4 upregulated, 4 downregulated) is not fully justified biologically or statistically. Were these the top DEGs by p-value, fold-change, or literature relevance? A more systematic, data-driven method, like using adjusted p-value alongside biological relevance, would be stronger. Authors should discuss why they have emphasized on these genes leaving other significantly differentially expressed genes.

4. While RT-qPCR validation on 17 more SA samples is good, the lack of stratification by arrest stage in this cohort is a weakness. Genes might behave differently and express differently across these stages. If authors could not do to rectify this, then at least they should consider it as their limitation.

B. Minor Issues

1. Figures are not clear, may be due to submission related deformation. Figure and Table legends should be more descriptive to allow understanding without the main text.

2. The term “spermatogenesis arrest” is sometimes abbreviated as SA and sometimes written out—consistency is advised.

4. Suggestions for Improvement

1. Clarify statistical thresholds and gene selection: Mention precise thresholds for DEG selection (adjusted p-value, log2FC) and how the final eight genes were shortlisted.

2. Discuss broader implications: Expand on how these candidate genes could be targeted or screened clinically, or how they relate to known pathways in spermatogenic regulation.

3. Line no 92 to 100: This part may not be required and seems to be irrelevant with the article. This portion can be omitted.

4. Materials and methods: Inclusion criteria regarding case sample selection should be more precise. Whether case patients were checked for Y chromosome microdeletion or not is to be confirmed. Y chromosome microdeletion is an important consideration as it can strongly influence the gene expression profile.

5. Quality of the figures should be enhanced.

Recommendation

Major Revision

The manuscript presents a well-executed and promising study on spermatogenesis arrest using transcriptomics. However, above mentioned issues are to be and must be addressed.

Reviewer #2: This is a technically sound and well-structured manuscript investigating the transcriptomic landscape of spermatogenesis arrest (SA) in non-obstructive azoospermia (NOA) patients. The use of high-throughput RNA-sequencing and validation by RT-qPCR on testicular tissue samples represents a valuable approach to better understand molecular mechanisms underlying male infertility.

The authors clearly define their objectives, and the methods are described in sufficient detail to allow reproducibility. The integration of bioinformatics analysis, including differential gene expression, GO and KEGG pathway enrichment, and PCA, is appropriate and strengthens the conclusions drawn. The results are consistent and well presented. The correlation between RNA-seq and RT-qPCR data is particularly convincing.

However, I have a few minor revisions to suggest:

Figures:

All figures currently lack descriptive titles or clear legends. Please provide informative captions for each figure, explaining what is shown, group comparisons, statistical indicators, and the meaning of colors or symbols used.

The quality of the figures is suboptimal. They appear blurry, which makes them difficult to interpret. Please upload high-resolution versions.

Supplementary Material:

Ensure that all supplementary tables referenced in the text (e.g., Table S1, S2, S3, S4) are available and properly labeled in the submission package.

Clarify the use of control data:

Since RNA-seq control samples were obtained from previously published datasets, please clearly state any differences in data acquisition or batch effects that may have occurred and how they were addressed in the analysis.

Gene validation rationale:

While the selection of top up/down-regulated genes is valid, a brief explanation for choosing these specific genes for RT-qPCR (e.g., biological relevance, novelty, expression intensity) would enhance clarity.

English Language and Formatting:

The manuscript is generally well written. However, a minor revision by a native English speaker or language editor would help improve fluency and consistency.

For example, correct terms like “SPRR2C, a pseudogene” → “a small proline-rich protein-2C, a pseudogene product involved in...”, or revise redundant phrases for conciseness.

Discussion:

The discussion is comprehensive and rich in references. However, a brief paragraph summarizing the clinical implications of the findings (e.g., potential diagnostic or therapeutic targets) at the end would strengthen the conclusion.

Overall, this is a solid and promising manuscript that contributes new insights to the understanding of testicular transcriptomics in male infertility. I recommend minor revision before acceptance.

**Do you want your identity to be public for this peer review?** For information about this choice, including consent withdrawal, please see our Privacy Policy

Reviewer #1: No

Reviewer #2: No

---

## [Author Response · Author response to Decision Letter 1]

31 Jul 2025

Please see the enclosed response file to find all of our responses to the reviewers' comments

---

## [Decision Letter · Decision Letter 1]

26 Aug 2025

Gene expression alterations in testicular biopsies

from males with spermatogenesis arrest identified by transcriptome analysis

PONE-D-25-14380R1

Dear Dr. Batiha,

We’re pleased to inform you that your manuscript has been judged scientifically suitable for publication and will be formally accepted for publication once it meets all outstanding technical requirements.

Kind regards,

Su-Ren Chen

Academic Editor

PLOS ONE

Additional Editor Comments (optional):

 I read the responses to reviewer 2 carefully. I can suggest publication at this version. There are still some spelling/grammatical mistakes in the revised manuscript and the authors should carefully correct them during proofreading

Please note that I have acted as a reviewer for this manuscript, and you will find my comments below, under Reviewer 3.

Reviewers' comments:

Reviewer's Responses to Questions

**Comments to the Author**

Reviewer #1: All comments have been addressed

Reviewer #3: (No Response)

2. Is the manuscript technically sound, and do the data support the conclusions?

Reviewer #1: Yes

Reviewer #3: (No Response)

3. Has the statistical analysis been performed appropriately and rigorously?

Reviewer #1: Yes

Reviewer #3: (No Response)

4. Have the authors made all data underlying the findings in their manuscript fully available?

Reviewer #1: Yes

Reviewer #3: (No Response)

5. Is the manuscript presented in an intelligible fashion and written in standard English?

Reviewer #1: Yes

Reviewer #3: (No Response)

Reviewer #1: The authors have provided satisfactory answer to all my queries from the previous version of the manuscript. In my opinion the manuscript is acceptable in its current form for publication.

Reviewer #3: I read the responses to reviewer 2 carefully. I can suggest publication at this version. There are still some mistakes in the revised manuscript and the authors should carefully correct them during proofreading. I also suggest the authors to make it clear why they chose one sample at the spermatogonial stage and four samples at the spermatid stage (genes should behave differently and express differently across these two types of spermatogeneic arrest). I agree with that the authors provide useful source of RNA-seq data from spermatogenic arrest patients; however, the claim of ‘potential diagnostic or therapeutic targets’ is talking in a vague and vague manner.

**Do you want your identity to be public for this peer review?** For information about this choice, including consent withdrawal, please see our Privacy Policy

Reviewer #1: **Yes: ** Sujay Ghosh

Reviewer #3: **Yes: ** Su-Ren Chen

---

## [Editor Report · Acceptance letter]

PONE-D-25-14380R1

PLOS ONE

Dear Dr. Batiha,

I'm pleased to inform you that your manuscript has been deemed suitable for publication in PLOS ONE. Congratulations! Your manuscript is now being handed over to our production team.

Kind regards,

on behalf of

Prof. Su-Ren Chen

Academic Editor

PLOS ONE